# The oldest magnetic record in our solar system identified using nanometric imaging and numerical modeling

Jay Shah[1,2,7], Wyn Williams[3], Trevor P. Almeida [1,4], Lesleis Nagy[3], Adrian R. Muxworthy[1], András Kovács[5], Miguel A. Valdez-Grijalva[1], Karl Fabian[6], Sara S. Russell[2], Matthew J. Genge [1] & Rafal E. Dunin-Borkowski [5]

Recordings of magnetic fields, thought to be crucial to our solar system's rapid accretion, are potentially retained in unaltered nanometric low-Ni kamacite (~ metallic Fe) grains encased within dusty olivine crystals, found in the chondrules of unequilibrated chondrites. However, most of these kamacite grains are magnetically non-uniform, so their ability to retain four-billion-year-old magnetic recordings cannot be estimated by previous theories, which assume only uniform magnetization. Here, we demonstrate that non-uniformly magnetized nano-metric kamacite grains are stable over solar system timescales and likely the primary carrier of remanence in dusty olivine. By performing in-situ temperature-dependent nanometric magnetic measurements using off-axis electron holography, we demonstrate the thermal stability of multi-vortex kamacite grains from the chondritic Bishunpur meteorite. Combined with numerical micromagnetic modeling, we determine the stability of the magnetization of these grains. Our study shows that dusty olivine kamacite grains are capable of retaining magnetic recordings from the accreting solar system.

[1] Department of Earth Science and Engineering, Imperial College London, London SW7 2BP, UK. [2] Department of Earth Sciences, Natural History Museum, London SW7 5BD, UK. [3] School of Geosciences, University of Edinburgh, Edinburgh EH8 9XP, UK. [4] School of Physics and Astronomy, University of Glasgow, Glasgow G12 8SU, UK. [5] Ernst Ruska-Centre for Microscopy and Spectroscopy with Electrons and Peter Grünberg Institute, Jülich 52425, Germany. [6] Geological Survey of Norway, Trondheim 7491, Norway. [7] Present address: Department of Earth, Atmospheric and Planetary Sciences, Massachusetts Institute of Technology, Cambridge, MA 02139, UK. Correspondence and requests for materials should be addressed to J.S. (email: jayshah@mit.edu)

Unaltered meteorites originating from our own proto-planetary disk acquired a thermoremanent magnetization (TRM) during formation and present an excellent opportunity to understand the extent of the early solar system magnetic field. The most likely material to have retained this field information is dusty olivine: assemblages of nanometric low-Ni kamacite grains protected from alteration by their host olivine crystal, found in the chondrules of unequilibrated primitive chondrites[1,2]. A recent estimate of the ancient magnetic field intensity (paleointensity) from dusty olivine in Semarkona[3] has provided an upper bound of 54 ± 21 µT for the magnetic field present in the chondrule-forming region (2.5 astronomical units (au)) of the protoplanetary disk during its first two to three million years[4,5]. This estimate is widely used in models for chondrule formation[6,7] and for the accretionary dynamics of the protoplanetary disk[8,9].

The magnetization carriers in dusty olivine are dominantly kamacite grains that have sizes greater than 25 nm and support non-uniform vortex magnetization states[10,11]. Retention of magnetic remanence over geological timescales, which is the underpinning hypothesis that enables paleomagnetism, is only predicted for uniformly magnetized grains by Néel's single domain (SD) theory[12]. Non-uniformly magnetized grains such as magnetic vortex states are not described by Néel's SD theory. Despite efforts to understand magnetic vortex states[13,14], it is unknown whether non-uniformly magnetized kamacite grains can retain their TRM for solar system timescales, i.e., 4.6 Ga. It is therefore of great importance to establish which magnetization states occur in the natural remanence carriers, and whether these non-uniform magnetization states can retain a magnetic remanence imparted by magnetic fields that were present in the protoplanetary disk billions of years ago[12,15].

Here we study chondrules from the unequilibrated ordinary chondrite Bishunpur (LL3.1) using the advanced transmission electron microscope (TEM) technique of in-situ temperature-dependent off-axis electron holography[16] (nanometric magnetic imaging) and numerical micromagnetic modeling[17] to determine whether dusty olivine can retain a record of the magnetic field from the early solar system.

## Results

**Room-temperature off-axis electron holography**. We recorded room-temperature magnetic induction maps from 19 kamacite grains (Fig. 1 and Supplementary Figure 1) using off-axis electron holography (hereafter holography) (see Methods) from the meteorite Bishunpur (LL3.1). Scanning TEM (STEM) energy dispersive X-ray spectroscopy analysis was used to establish that the kamacite grains are almost pure Fe and are encased in forsteritic olivine (see Supplementary Figure 2). The average axial ratio (AR; length/width) of the dusty olivine kamacite grains is 1.5, they are ~ 150–600 nm in size (average 353 ± 137 nm × 250 ± 106 nm), and are typically found to have well-defined single vortex (SV) magnetization states with their vortex cores aligned out-of-plane and with little external stray magnetic fields (Fig. 1 and Supplementary Figure 1). Our findings are in accordance with previous holography analyses of dusty olivine[10,11].

**Temperature-dependent off-axis electron holography**. We recorded in-situ temperature-dependent holographic magnetic induction maps (see Methods) of four kamacite grains and present the heating sequence for one of them in Fig. 2. The representative kamacite grain shown in Fig. 2 was focused ion beam (FIB) milled from its original morphology until it was electron transparent for in-situ TEM experiments, likely affecting its AR. Its saturated remanent magnetization state, which was induced at

room temperature, resembles that of a uniformly magnetized grain or an in-plane vortex-core magnetization (Fig. 2b). This remanent state was maintained when the grain was heated to 500 °C, with little change in its direction or intensity (Fig. 2b–g). At 600 °C, the grain underwent chemical alteration (Supplementary Figure 3), likely through a reaction with the surrounding olivine, as the TEM operates in high vacuum. Chemical alteration prevents accurate determination of the magnetization state beyond 600 °C, due to the difficulty of removing the mean inner potential contribution to the phase recorded from the new mineralization.

**High-temperature micromagnetic modeling of a large grain**. In order to determine whether the 458 × 98 × 60 nm grain was in a uniform or a vortex state, we used a finite element method (FEM) micromagnetic algorithm[17] (MERRILL, see Methods) to model the three-dimensional magnetization states compatible with the grain's shape and mineralogy. We found that the grain was in a multi-vortex state with its magnetization aligned with the long axis (also the saturation axis) (Fig. 2h, i). Using a nudged elastic band (NEB) numerical algorithm[17–20], we then calculated the energy barriers related to changes of the magnetization state. The thermal relaxation time across these barriers at 300 °C, the highest temperature reached by Bishunpur chondrules since formation 4.6 Ga[21], are many orders of magnitudes longer than the age of the solar system (see Supplementary Note 1–3 and Supplementary Figures 4–9).

**Micromagnetic modeling of Fe parallelepipeds**. In order to determine the stability of dusty olivine kamacite grains in more general cases, we used the MERRILL path minimization algorithm[17–20,22] to calculate the thermal relaxation times as a function of size and AR for Fe cubes and cuboids. Initially, we found local-energy minimum (LEM) magnetization states for the Fe cubes and cuboids by performing 100 energy minimizations for randomized magnetization directions for each morphology (Fig. 3). For the smaller grain sizes (below 23 nm), the LEM states correspond to uniform magnetization states that are aligned with the easy magnetocrystalline axis for equidimensional grains and with the long axis for elongated grains (Fig. 3a). As the grain size increases toward 23 nm, there is increased "flowering"[23,24] (Fig. 3a). In equidimensional Fe grains that have sizes above 23 nm, magnetic vortex states with their cores aligned along the hard

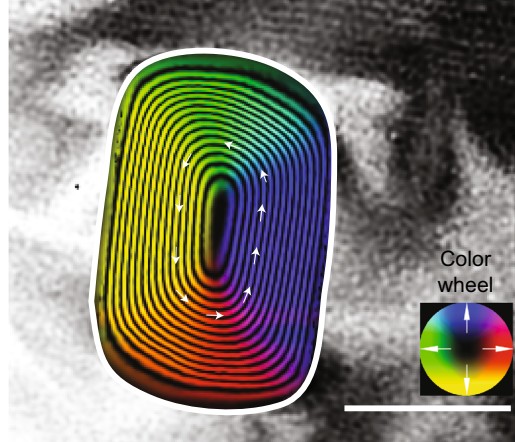

**Fig. 1** Magnetic induction map of a Bishunpur kamacite grain. Magnetic induction map of a kamacite grain in dusty olivine reconstructed from electron holograms acquired at room temperature. The contour spacing is π radians. The direction of the projected in-plane magnetic induction is indicated by the arrows and the color wheel. Scale bar: 200 nm

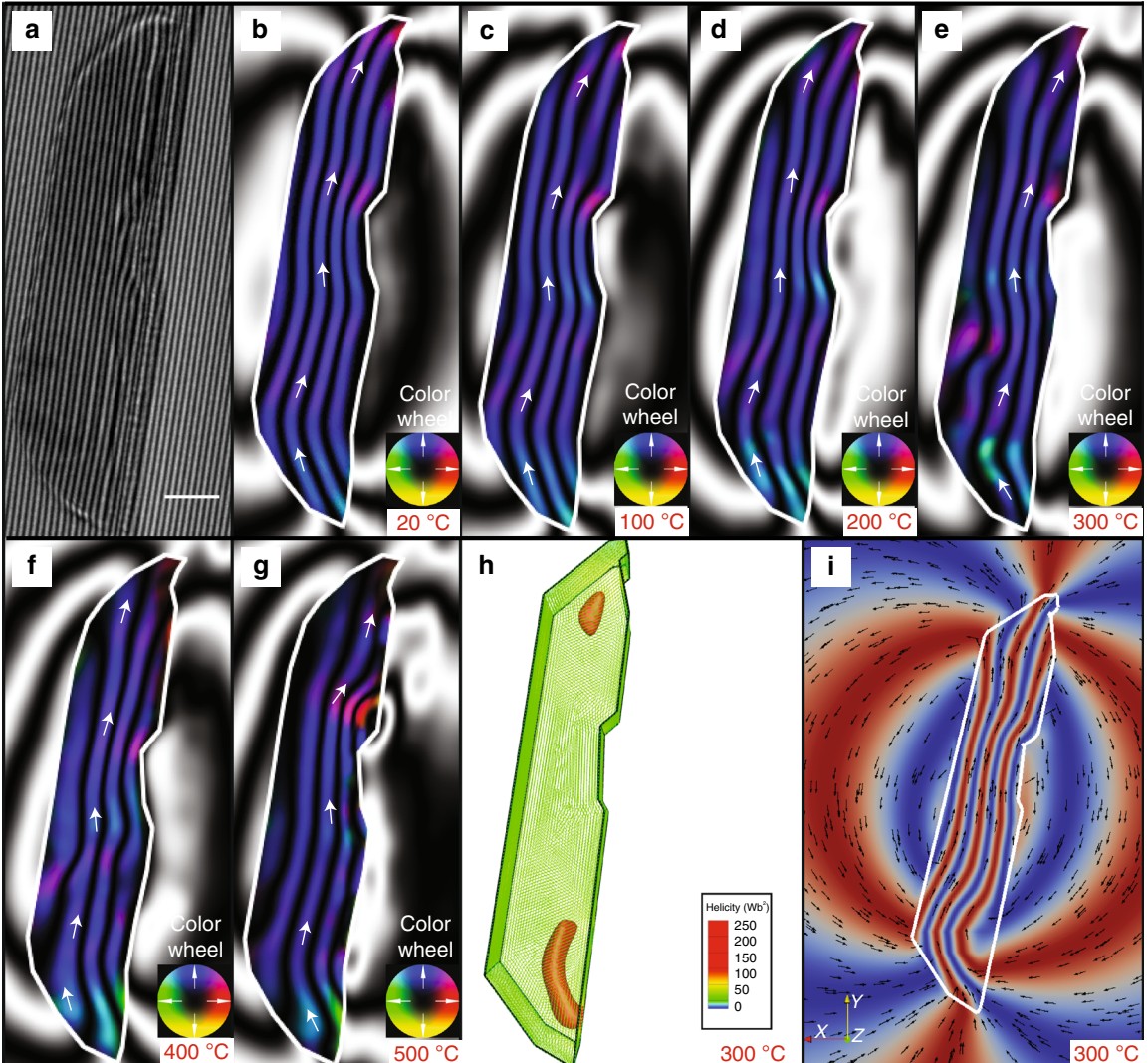

**Fig. 2** Visualizing the magnetization of a Bishunpur kamacite grain during in-situ heating. **a** Electron hologram of a Bishunpur dusty olivine kamacite grain before heating. The fringes due to electron beam interference are used to determine the phase shift of the electron beam passing through the sample. **b–g** Magnetic induction maps reconstructed from holograms of the kamacite grain heated in-situ from 20 to 500 °C (the grain underwent chemical alteration above 500 °C). **h** Micromagnetic model of a minimum energy state magnetization distribution at 300 °C for a Fe tetrahedral mesh modeled after the grain in **a–g**. Curie temperature for Fe is 760 °C and peak temperature of Bishunpur is ~ 300 °C[21]. The regions of high helicity (red) are highlighted by a contour plot to display the vortex cores in the modeled kamacite grain. **i** Electron holography-style magnetic induction map simulated from the micromagnetic solution in **h**. The contour spacing for **b–g** is 1.57 radians. The direction of the projected in-plane magnetic induction in **b–g** is indicated by the arrows and the color wheel. Scale bar: 50 nm

magnetocrystalline axis are the LEM state (Fig. 3b), whereas for sizes above 27 nm the core aligns with the easy magnetocrystalline axis (Fig. 3c). In elongated Fe grains, the core aligns with the short axis.

Transition paths between vortex LEM states were found to be structure-coherent rotations[22] of the vortex core from one LEM state to another (see Supplementary Movie 1), in agreement with previous observations of magnetite[22]. Although the individual moments do not rotate coherently, the rotation of the vortex core itself is similar to the coherent rotation of magnetization vectors that we observe in uniform LEM states. The energy barriers between uniform states (Fig. 4) are very low for equidimensional Fe grains (Fig. 3a). Equidimensional grains that have sizes below 29 nm are unstable on solar system timescales, as all uniform SD magnetization states in equidimensional grains are unstable over this timescale, although different reversal modes are active at different grain sizes[13]. Astonishingly, for equidimensional cubes

only vortex states with their cores aligned along easy axes in grains with sizes above 29 nm are capable of retaining magnetizations over solar system timescales. We found that these states are stable up to at least grain sizes of 50 nm, which was the largest SV modeled.

Elongation of the grain increases the stability of the magnetization state and increases the uniform to non-uniform transition size (Fig. 4)[25]. Uniform magnetization states increase in energy barrier with increasing size, whereby a flowering of the magnetization vectors at the grain edges further increases the structural stability up to a peak close to 20 nm (Fig. 4). Beyond this peak, a vortex is formed during the magnetization reversal, which leads to an intermediate decrease in the energy barrier with increasing grain size[13] up to a trough at 25–35 nm (Fig. 4). For larger grain sizes, the easy axis vortex state is the LEM, increasing in stability with increasing grain size (Fig. 4). The kamacite grains that are found in chondrules from Semarkona[11]

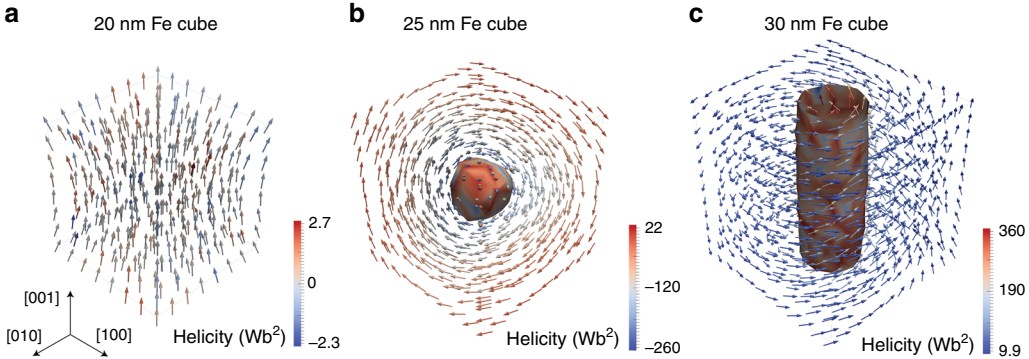

**Fig. 3** Representative magnetization states for Fe cubes. Global energy minimum (GEM) magnetization states for cubes of Fe determined using a finite element method (FEM) micromagnetic algorithm. **a** 20 nm cube in a uniform magnetization state along the easy axis; **b** 25 nm cube in non-uniform hard axis magnetization state; **c** 30 nm cube in a non-uniform easy axis magnetization state. Helicity was determined by calculating $\mathbf{m} \cdot (\nabla \times \mathbf{m})$, where $\mathbf{m}$ is the magnetization vector. Regions of high (±) helicity are highlighted by a contour plot to display vortex cores, e.g., in **b** and **c**

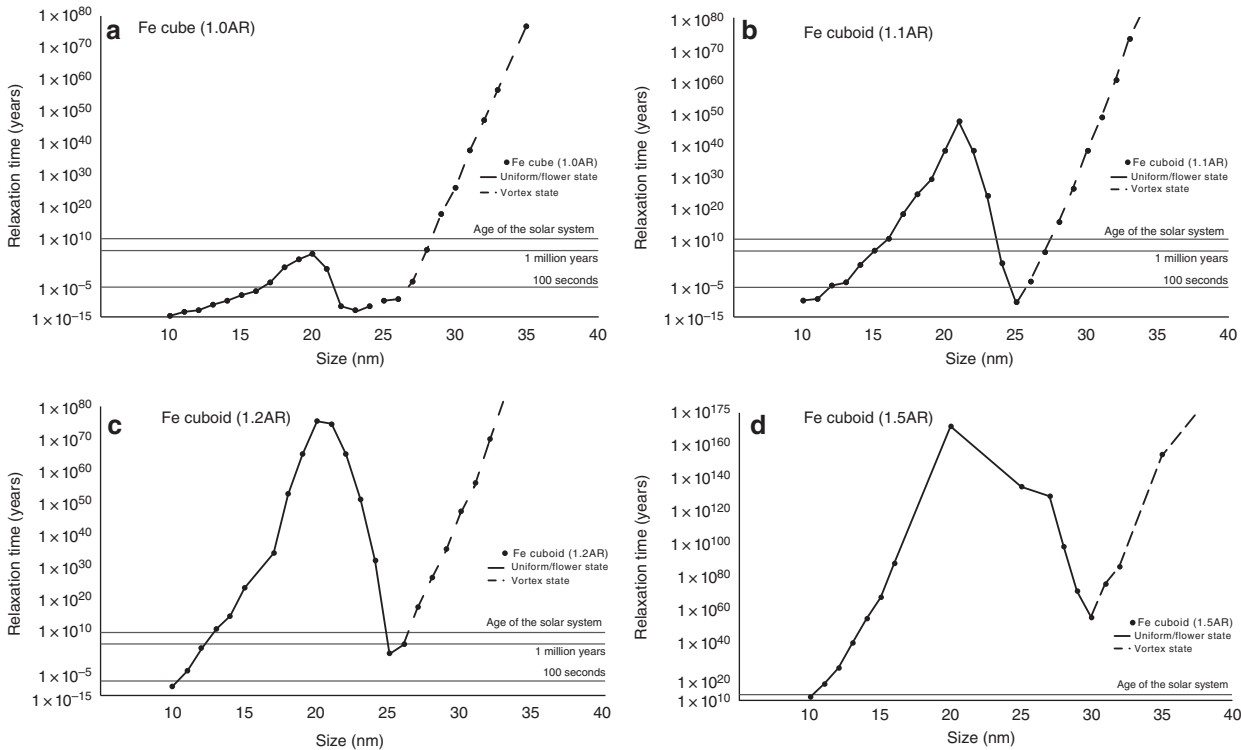

**Fig. 4** Relaxation times for Fe cubes and cuboids at room temperature. **a–d** Graph displaying the relaxation time, i.e., stability for different values of axial ratio (AR) and size of Fe cuboids calculated using the nudged elastic band (NEB) method for global energy minimum (GEM) magnetization states determined using a finite element method (FEM) numerical algorithm. The size describes the volume of the equivalent cube (i.e., the volume of a 20 nm cuboid with a 1.1, 1.2, or 1.5 AR is equivalent to the volume of a 20 nm cube). Horizontal lines mark relaxation times of 100 s, 1 Ma, and the age of the solar system (4.6 Ga). Solid versus dashed curves indicate whether the GEM state is a uniform or vortex magnetization state. Incoherent rotation of uniform states, which proceeds via intermediate vortex states results in a reduction in stability. When the GEM assumes a non-uniform vortex state (dashed curves) through <100>, its stability increases with increasing grain size. Observed chondrule dusty olivine kamacite grains have an average AR of 1.5, which according to the above data are stable over solar system timescales irrespective of magnetization state for all of the sizes tested (10–50 nm)

and Bishunpur (this study) have ARs of ~ 1.5. At such elongations for all grain sizes modeled (10–50 nm) the magnetizations are stable for timescales greater than the age of the solar system, independent of their uniform or non-uniform states (Fig. 4d). The lower temperatures that are experienced in space only slightly change the Fe material parameters, but significantly decrease thermal activation, and thus increase the calculated relaxation times. Therefore, micromagnetic modeling strongly indicates that the kamacite TRM imparted during dusty

olivine formation in the protoplanetary disk remains stable to the present day (Fig. 4).

Furthermore, the remanence imparted during dusty olivine formation would have survived the heating that Bishunpur is predicted to have experienced since its accretion (~ 300 °C)[21]. Temperature-dependent electron holography reveals for the first time the high-temperature stability of non-uniform remanent magnetization states in low-Ni kamacite directly observed up to 500 °C and the obtained thermal relaxation times at 300 °C are

longer than the age of the solar system (Fig. 2b–g and Supplementary Notes 1–3). This confirms that even multi-vortex states can carry a primary remanent magnetization from the protoplanetary disk.

## Discussion

Paleomagnetic data are some of the only sources of evidence of early solar system conditions that constrain mechanisms of heating and momentum transport in the protoplanetary disk[6–9,26,27]. Our observations and calculations show that SV or multi-vortex magnetization state Fe grains in dusty olivine will carry magnetic remanence originating from the early solar system. Most current paleointensity protocols implicitly assume that the magnetization carriers behave like uniform SD magnetization states, as the protocols are based on Néel's theory of SD grains[12]. Non-uniform magnetization states are the most abundant state of magnetization present in rocks and meteorites, however their thermal and temporal stabilities are poorly understood and they have previously been considered to be poor magnetic recorders. This study presents a step change in our understanding of non-uniform magnetic states. It is now clear that a more comprehensive understanding of the thermomagnetic characteristics of magnetic vortex states will facilitate more sophisticated and sample-specific paleointensity estimates, which will further our understanding of how the protoplanetary disk evolved into our present-day planetary system.

## Methods

**Sample preparation for electron microscopy.** Samples for the advanced TEM technique of off-axis electron holography (hereafter holography) were prepared using FIB milling from a polished section of the Bishunpur meteorite and either attached to a Cu Omniprobe grid for room temperature analysis or placed on the windows of a silicon nitride EMheaterchip for in-situ heating in a DENSSolutions double tilt TEM specimen holder. FIB milling, (S)TEM imaging, chemical analysis, and holography experiments were conducted at the Ernst-Ruska Centre for Microscopy and Spectroscopy with Electrons, Forschungszentrum Jülich, Germany.

**Off-axis electron holography.** Electron holograms were acquired using an FEI Titan 80–300 TEM operated in Lorentz mode at 300 kV using a charge-coupled device camera and an electron biprism typically at 50 V. Magnetic induction maps were recorded after tilting the sample to ± 70° and applying a vertical magnetic field of > 1.5 T using the conventional microscope objective lens, in order to acquire images before and after reversing the direction of magnetization in the sample. Evaluation of half of the difference between phase images recorded with opposite magnetization directions in the sample was used to remove the mean inner potential contribution to the phase. The mean inner potential was subtracted from the unwrapped total phase shift in order to construct magnetic induction maps that were representative of the magnetic remanence[28].

**Temperature-dependent electron holography.** In order to determine the change in magnetic induction during heating, the sample was magnetized and images were recorded at room temperature, at 100 °C, and then at temperatures up to 800 °C in 100 °C intervals. The same procedure was followed during cooling. The ramp during heating was 50 °C min$^{-1}$ and each temperature interval was maintained for 10 min, to allow sufficient time for imaging. The mean inner potential was subtracted from the unwrapped total phase shift acquired at each temperature interval, to allow the construction of magnetic induction maps representative of the magnetic remanence, as shown previously[29].

**Micromagnetic modeling.** Magnetic domain stability is highly grain-size-dependent. At very small grain sizes, uniform magnetization is typically unstable due to thermal fluctuations. As the grain size increases, a non-uniform magnetization state becomes the most energetically favorable state[23,25]. We determined the magnetization states associated with different grain sizes of Fe using FEM micromagnetic simulations[17]. Tetrahedral meshes were generated for this using MRshRRILL and FEM models were performed using MERRILL[17]. The magnetic free energy was determined for each of the tetrahedra and summed over all tetrahedra to determine $E_{tot}$, which the FEM discretized for the minimization of an initial state, **m**, where the magnetization at each node of each element was given a

random direction for the grain in question, $\Omega$, according to the expression

$$E_{tot} = \int_{\Omega} \left[ A|\nabla \mathbf{m}| + K_1 \left[ m_x^2 m_y^2 + m_x^2 m_z^2 + m_y^2 m_z^2 \right] - M_s[\mathbf{H}_z \cdot \mathbf{m}] - \frac{M_s}{2}[\mathbf{H}_d \cdot \mathbf{m}] \right] dV,$$

(1)

where the material is defined by the following temperature-dependent parameters: $A$, the exchange constant; $K_1$, the magnetocrystalline anisotropy; and $M_s$, the saturation magnetization. $\mathbf{H}_z$ and $\mathbf{H}_d$ are external and self-demagnetizing fields, respectively. The material parameter constants used for room temperature Fe[25]: $A = 2 \times 10^{-11}$ Jm$^{-1}$, $K_1 = 4.8 \times 10^4$ Jm$^{-3}$, and $M_s = 1.72 \times 10^6$ Am$^{-1}$. The material parameter constants used for Fe at 300 °C: $A = 1.52 \times 10^{-11}$ Jm$^{-1}$, $K_1 = 2.2 \times 10^4$ Jm$^{-3}$, and $M_s = 1.61 \times 10^6$ Am$^{-1}$.

LEM magnetization states are found by minimizing $E_{tot}$ using a modified conjugate gradient method[18]. For each grain geometry and size for which the relaxation time was evaluated, 100 minimizations were performed to calculate the most favorable LEM states. Two different magnetization states, $L_1$ and $L_2$, with lowest energy were then selected as the start and end configurations of an initial path of 100 magnetization states transforming $L_1$ into $L_2$. MERRILL's combined NEB and action minimization method was used to determine the nearest minimum-action path connecting $L_1$ and $L_2$, which also defines the corresponding thermal energy barrier[17,18]. For non-uniform vortex states, $L_1$ and $L_2$ were required to have the same helical sense of vortex core rotation (helicity), as unwinding of the core requires much more energy than retaining the same helicity. Helicity was determined by calculating $\mathbf{m} \cdot (\nabla \times \mathbf{m})$, where $\mathbf{m}$ is the magnetization vector. The relaxation time ($\tau$) is related to the energy barrier ($\Delta E$) by the Néel–Arrhenius equation[12]

$$\tau = \frac{1}{C} e^{\Delta E / k_B T}$$

(2)

where $C$ is the atomic switching frequency ($10^{-9}$ s), $k_B$ is Boltzmann's constant, $T$ is the temperature in Kelvin, and $\Delta E = E_S - E(L_1)$ is the energy difference between the highest saddle point and the LEM $L_1$ determined by the NEB method. The relaxation time directly determines whether dusty olivine can theoretically retain its magnetization over solar system timescales.

**Data availability.** The data that support the findings of this study are available from the corresponding author upon request.

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

## Acknowledgements

We thank D. Meertens for FIB lamellae preparation. We thank Pádraig Ó Conbhuí for assistance with MERRILL and for writing MEshRILL, which was used to create the meshes used in this study. This work was funded by the STFC (grant number ST/N000803/1). Meteorites were provided by the Natural History Museum, London. The research leading to these results has received funding from the European Research Council under the European Union's Seventh Framework Programme (FP7/2007-2013)/ERC grant agreement number 320832.

## Author contributions

J.S. conducted the room-temperature and temperature-dependent off-axis electron holography experiments, the micromagnetic simulations of the 10–50 nm Fe grains, wrote the manuscript, and produced Figs. 1–4, Supplementary Figures 1–3, and Supplementary Movie 1. W.W. and L.N. conducted the high-temperature micromagnetic simulations of the Bishunpur kamacite grain. W.W. wrote Supplementary Notes 1–3 and produced Supplementary Figures 4–9. J.S. analyzed the room-temperature electron holography data and T.P.A. analyzed the temperature-dependent electron holography data. T.P.A. and A.K. assisted with electron holography experiments and analysis. M.V.-G. wrote bash scripts that helped to streamline the numerical analysis and assisted with the presentation of the numerical data. W.W. and K.F. wrote the micromagnetic code MERRILL used for numerical analysis. J.S., A.R.M., S.S.R., and M.J.G. discussed the results and the petrology. A.R.M. had the original idea for the study, led the direction of the study, and helped to write the manuscript. All authors discussed and commented on the results and the manuscript.

## Additional information

**Competing interests:** The authors declare no competing interests.

