## [Peer Review File(PDF 292 kb) · Nature Communications]

Reviewers' comments:

Reviewer #1 (Remarks to the Author):

Reply to Author about Q1

I had completely misread the manuscript. Yes, their calculations truly showed that the relaxation time of their coarse-grained vortex kamacites are many order of magnitudes longer than the age of 100 million years metamorphism at 300°C. This means that the thermal remanence of Bishunpur chondrules retains its remanence since their formation. I agree with them. However, authors should describe the influence of subsequent metamorphism (Rambaldi and Wasson (1981) in order not to mislead readers.

Reply to Author about Q2

Even non-uniform vortex grains retain their remanence since the formation, there is no vital link to astrophysical processes to give it the high-impact takeaway message in the manuscript that is necessary for Nature Communication. I still believe that authors could challenge to estimate paleointensity from their thermoremanent magnetization experiments, or to give the readers a hint to estimate paleointensity from multi-vortex kamacite grains from the chondritic Bishunpur meteorite. I believe that present manuscript is still for specialists of paleomagnetism.

Overall, this manuscript is ready for the acceptance if authors add the description about the high-impact takeaway message about solar system formation.

Reviewer #2 (Remarks to the Author):

The paper under review aims to investigate the paleomagnetic fidelity of Fe particles within dusty olivine grains from unequilibrated ordinary chondrites. The authors have achieved this by employing innovative experimental measurements and novel calculations. This paper was originally submitted to Nature Astronomy and my review of the current submission to Nature Communications will focus specifically on the comments I gave to the first submission as the resubmitted version of the paper has only received minor modifications since the first submission. My review of the original paper focussed on 5 main aspects:

- 1) Whether the impact of the paper was high enough for Nature Astronomy
- 2) The novelty of the measurements
- 3) The size range of the particles in the models
- 4) The possible inclusion of bulk measurements
- 5) The conclusion that the results of this paper open previous paleointensity estimates from dusty olivines to debate.

In the author's rebuttal to the reviews of the initial submission, they successfully demonstrated that they did in fact run models with particle sizes matching those observed experimentally and make a convincing argument that bulk measurements would be difficult to make. I am happy to accept these rebuttals and I am glad to see that a size range has been added to the main text to facilitate understanding and state explicitly that they have modelled particles with a size that matches those observed experimentally. I am also happy to accept that previous thermal electron holography measurements were made on magnetite and, as the study under review focusses on Fe particles within natural dusty olivines, the results are not a simple reproduction of previous studies and have an element of novelty.

However, I still have major and significant concerns over the final paragraph of the paper. The authors have still included the statement that previous 'paleofield estimates are open to debate' because they only observed nonuniform, vortex domain states in the Fe particles in dusty olivine.

In my previous review, I argued that this statement is unsupported by the data the authors collected, and the authors appear to agree with me in their rebuttal by stating that 'An exhaustive database of large grains at variable temperatures is required to fully understand the thermomagnetic properties of vortex state grains before a quantitative calibration of paleointensity estimates can be made.' It appears as though the authors are saying that they do not have the data to successfully and reliably recalculate paleointensity estimates using an updated calibration factor and therefore they have no way of quantitatively demonstrating that the paleointensities inferred from the previous study are in fact open to debate. In this reviewer's opinion, suggesting that these paleointensity estimates are 'open to debate' implies that the actual paleointensities differ significantly (i.e., outside of the error range) from those previously reported. In my opinion, the authors cannot say that the actual paleointensities would, in fact, be outside of the error range of previous paleointensity estimates as they have not obtained an updated calibration factor to recalculate these values. Consequently, including such a statement without supporting evidence, acts to cast an unnecessary and unwarranted shadow of doubt on previously collected data. Given the important and inter-disciplinary nature of paleointensity estimates of the nebula field, it should be encouraged that unsupported statements such as these are not published as they act to unnecessarily confuse researchers who rely on these results but do not know the details of paleo- and rock magnetic measurements.

Elaborating on this point, Lappe et al., *G3*, Q12Z35, (2011) performed the first calibration of dusty olivine using synthetic samples. Although the size of the Fe particles in these synthetic samples is admittedly smaller than those observed in natural dusty olivines, these authors observe a majority of Fe particles with magnetic vortices, while also observing a minority with a uniform single-domain state. In fact, Einsle et al., *Am Min*, 101, 2070-2084, (2016) state that 'Lappe et al. (2011) similarly identified a dominance of single vortex states, but noted also a significant number of SD particles.' Recent FORC analysis (Lappe et al., PhD Thesis, and Harrison et al., International Conference on Rock Magnetism 2017) further support the claim that the dominant carrier in these synthetic samples are vortices. The existence of a minor proportion of single domain particles in the synthetic samples used by Lappe et al. (2011) will result in a different calibration factor than that that would be obtained from natural samples. However, the dominant carrier in both the synthetic samples and the natural samples (as identified by the paper under review) appears to be vortices, so in all likelihood, the calibration factor identified by Lappe et al. (2011) will be similar to the calibration factor that would be inferred from natural samples. Consequently, the paleointensities inferred from the synthetic and natural calibration factors will be similar. Importantly, the authors of the paper under review do not present a new calibration factor calculated from natural samples or quantitatively demonstrate how much this new factor would change previous paleointensity estimates. Therefore, there is no quantitative demonstration in the paper under review that the complete dominance of vortices will cause the paleointensity estimates to fall outside of the 54 ± 21 μT error range identified by Fu et al., *Science*, 346, 1089-1092, (2014) from natural samples using the calibration factors obtained by Lappe et al. (2011) on their synthetic samples. So the statement in the paper under review that the previous paleointensities are 'open to debate' is unsupported by their data and is potentially misleading.

To summarise, the paper under review does not quantitatively constrain the effect of complete vortex dominance on paleointensity estimates from natural dusty olivines. Given the apparent dominance of vortices in both synthetic and natural samples, it is perfectly feasible that the paleointensities recovered from natural samples using an updated calibration factor will not differ significantly from the previously published values. Such a change would not have a significant effect on our overall qualitative understanding of the evolution of the early solar system. Saying that a measurement is 'open to debate' suggests that it may be significantly wrong, i.e., that the corrected value is outside of the error range of the previously suggested value and we would, therefore, have a new value of the nebula field paleointensity and we would have updated our understanding of the early solar system. As this cannot be proved to be the case without the quantitative argument that the actual paleointensity of the nebula field falls outside the error range of the previously published paleointensities, the statement that the previously published

paleointensities are 'open to debate' cannot be published.

The modelling and experimental work in this paper are potentially novel and significant enough to be published in Nature Communications even without the conclusion that challenges previous measurements. This paper represents a major experimental and modelling achievement, and the authors should be recognised for that. However, the conclusions of this paper as they read at the moment should not be published. The authors either need to calculate a new calibration factor and come up with updated paleointensity estimates (which they suggest in their rebuttal is outside the scope of their study) or come up with a convincing numerical argument as to why their current data support a paleointensity that is outside the error range of the previously reported value or change their text to reflect the fact that the complete dominance of vortices they observe will in all likelihood change the calibration factor and the recovered paleointensities, but that they cannot quantitatively or reliably say that this updated factor will cause the paleointensity estimates to differ significantly (i.e., outside of the error range) from previous estimates.

We thank you for taking the time to consider our manuscript and the reviewers for their useful comments and suggestions. We have responded to the reviews below.

Reviewers' comments:

Reviewer #1 (Remarks to the Author):

Reply to Author about Q1

I had completely misread the manuscript. Yes, their calculations truly showed that the relaxation time of their coarse-grained vortex kamacites are many order of magnitudes longer than the age of 100 million years metamorphism at 300°C. This means that the thermal remanence of Bishunpur chondrules retains its remanence since their formation. I agree with them. However, authors should describe the influence of subsequent metamorphism (Rambaldi and Wasson (1981) in order not to mislead readers.

Reply to Q1:

There is evidence presented by Rambaldi and Wasson (1981) to suggest that Bishunpur may have been heated up to 300-350°C and potentially to 400°C. Given that the relaxation time for the large grain calculated at 300°C is 3.74×10^{174} billion years, this increase in temperature is unlikely to result in a significantly shorter relaxation time.

Reply to Author about Q2

Even non-uniform vortex grains retain their remanence since the formation, there is no vital link to astrophysical processes to give it the high-impact takeaway message in the manuscript that is necessary for Nature Communication. I still believe that authors could challenge to estimate paleointensity from their thermoremanent magnetization experiments, or to give the readers a hint to estimate paleointensity from multi-vortex kamacite grains from the chondritic Bishunpur meteorite. I believe that present manuscript is still for specialists of paleomagnetism. Overall, this manuscript is ready for the acceptance if authors add the description about the high-impact takeaway message about solar system formation.

Reply to Q2:

Given that the stability and credibility of paleomagnetic data is often doubted by meteoriticists, as it is rarely challenged, our study is likely to be of interest to the wider meteoritical community, as well as to others who use paleomagnetic data.

We did not attempt or present any thermoremanent magnetization experiments. The sample preparation for the TEM experimental protocol conducted requires the application of strong magnetic fields, which precludes the retention of original magnetic data. The magnetizations analysed for stability were imparted in the laboratory. The point we wish to raise in our manuscript is that vortex magnetization states are stable and ubiquitous, yet largely ignored in analyses. We are now making significant steps to develop our understanding of them, which can lead to more accurate and robust paleointensity estimates provided that we continue to model and observe their behavior over a large range of shapes, sizes, and temperatures. Such a study is well beyond the scope of our present paper, and is the focus of a new grant application.

Reviewer #2 (Remarks to the Author):

Q1:

The paper under review aims to investigate the paleomagnetic fidelity of Fe particles within dusty olivine grains from unequilibrated ordinary chondrites. The authors have achieved this by employing innovative experimental measurements and novel calculations. This paper was originally submitted to Nature Astronomy and my review of the current submission to Nature Communications will focus specifically on the comments I gave to the first submission as the resubmitted version of the paper has only received minor modifications since the first submission. My review of the original paper focussed on 5 main aspects:

- 1) Whether the impact of the paper was high enough for Nature Astronomy*
- 2) The novelty of the measurements*
- 3) The size range of the particles in the models*
- 4) The possible inclusion of bulk measurements*
- 5) The conclusion that the results of this paper open previous paleointensity estimates from dusty olivines to debate.*

In the author's rebuttal to the reviews of the initial submission, they successfully demonstrated that they did in fact run models with particle sizes matching those obscured experimentally and make a convincing argument that bulk measurements would be difficult to make. I am happy to accept these rebuttals and I am glad to see that a size range has been added to the main text to facilitate understanding and state explicitly that they have modelled particles with a size that matches those observed experimentally. I am also happy to accept that previous thermal electron holography measurements were made on magnetite and, as the study under review focusses on Fe particles within natural dusty olivines, the results are not a simple reproduction of previous studies and have an element of novelty.

Reply to Q1:

We thank you for the useful comments and suggestions in the previous review and for accepting the rebuttals to the previous review.

Q2:

However, I still have major and significant concerns over the final paragraph of the paper. The authors have still included the statement that previous 'paleofield estimates are open to debate' because they only observed nonuniform, vortex domain states in the Fe particles in dusty olivine. In my previous review, I argued that this statement is unsupported by the data the authors collected, and the authors appear to agree with me in their rebuttal by stating that 'An exhaustive database of large grains at variable temperatures is required to fully understand the thermomagnetic properties of vortex state grains before a quantitative calibration of paleointensity estimates can be made.' It appears as though the authors are saying

that they do not have the data to successfully and reliably recalculate paleointensity estimates using an updated calibration factor and therefore they have no way of quantitatively demonstrating that the paleointensities inferred from the previous study are in fact open to debate. In this reviewer's opinion, suggesting that these paleointensity estimates are 'open to debate' implies that the actual paleointensities differ significantly (i.e., outside of the error range) from those previously reported. In my opinion, the authors cannot say that the actual paleointensities would, in fact, be outside of the error range of previous paleointensity estimates as they have not obtained an updated calibration factor to recalculate these values. Consequently, including such a statement without supporting evidence, acts to cast an unnecessary and unwarranted shadow of doubt on previously collected data. Given the important and inter-disciplinary nature of paleointensity estimates of the nebula field, it should be encouraged that unsupported statements such as these are not published as they act to unnecessarily confuse researchers who rely on these results but do not know the details of paleo- and rock magnetic measurements.

Elaborating on this point, Lappe et al., G3, Q12Z35, (2011) performed the first calibration of dusty olivine using synthetic samples. Although the size of the Fe particles in these synthetic samples is admittedly smaller than those observed in natural dusty olivines, these authors observe a majority of Fe particles with magnetic vortices, while also observing a minority with a uniform single-domain state. In fact, Einsle et al., Am Min, 101, 2070-2084, (2016) state that 'Lappe et al. (2011) similarly identified a dominance of single vortex states, but noted also a significant number of SD particles.' Recent FORC analysis (Lappe et al., PhD Thesis, and Harrison et al., International Conference on Rock Magnetism 2017) further support the claim that the dominant carrier in these synthetic samples are vortices. The existence of a minor proportion of single domain particles in the synthetic samples used by Lappe et al. (2011) will result in a different calibration factor than that that would be obtained from natural samples. However, the dominant carrier in both the synthetic samples and the natural samples (as identified by the paper under review) appears to be vortices, so in all likelihood, the calibration factor identified by Lappe et al. (2011) will be similar to the calibration factor that would be inferred from natural samples. Consequently, the paleointensities inferred from the synthetic and natural calibration factors will be similar. Importantly, the authors of the paper under review do not present a new calibration factor calculated from natural samples or quantitatively demonstrate how much this new factor would change previous paleointensity estimates. Therefore, there is no quantitative demonstration in the paper under review that the complete dominance of vortices will cause the paleointensity estimates to fall outside of the 54 ± 21 μT error range identified by Fu et al., Science, 346, 1089-1092, (2014) from natural samples using the calibration factors obtained by Lappe et al. (2011) on their synthetic samples. So the statement in the paper under review that the previous paleointensities are 'open to debate' is unsupported by their data and is potentially misleading.

To summarise, the paper under review does not quantitatively constrain the effect of complete vortex dominance on paleointensity estimates from natural dusty olivines. Given the apparent dominance of vortices in both synthetic and natural samples, it is perfectly feasible that the paleointensities recovered from natural samples using an updated calibration factor will not differ significantly from the previously published values. Such a change would not have a significant effect on our overall qualitative understanding of the evolution of the early solar system. Saying that a measurement is 'open to debate' suggests that it may be significantly wrong, i.e., that the corrected value is outside of the error range of the previously suggested value and we would, therefore, have a new value of the nebula field paleointensity and we would have the

update our understanding of the early solar system. As this cannot be proved to be the case without the quantitative argument that the actual paleointensity of the nebula field falls outside the error range of the previously published paleointensities, the statement that the previously published paleointensities are 'open to debate' cannot be published.

The modelling and experimental work in this paper are potentially novel and significant enough to be published in Nature Communications even without the conclusion that challenges previous measurements. This paper represents a major experimental and modelling achievement, and the authors should be recognised for that. However, the conclusions of this paper as they read at the moment should not be published. The authors either need to calculate a new calibration factor and come up with updated paleointensity estimates (which they suggest in their rebuttal is outside the scope of their study) or come up with a convincing numerical argument as to why their current data support a paleointensity that is outside the error range of the previously reported value or change their text to reflect the fact that the complete dominance of vortices they observe will in all likelihood change the calibration factor and the recovered paleointensities, but that they cannot quantitatively or reliably say that this updated factor will cause the paleointensity estimates to differ significantly (i.e., outside of the error range) from previous estimates.

Reply to Q2:

We agree that we are not yet able to make a quantitative calibration of paleointensity estimates. The interdisciplinary usage of paleomagnetic data highlights the importance of a study demonstrating the stability of magnetic remanence to the wider scientific community.

Calibration factors for paleointensity estimation by the ARM method have seen variation of up to an order of magnitude depending on the size and concentration of magnetic grains in tested samples (Banerjee and Mellema, 1974; Sugiura, 1979; Hartstra, 1982, 1983; Dunlop and Argyle, 1997; Peters and Dekkers, 2003; Yu, 2010; Lerner et al., 2017). Lerner et al. (2017) report a comparison and calibration of non-heating paleointensity methods, finding that the error of paleointensity estimates could be up to 50%. The calibration constant used by Fu et al. (2014) was determined by Lappe et al. (2013), however the range of bias fields used to determine this calibration constant (200-1500 μT) may well have introduced error to the calibration constant due to non-linear field dependencies of ARM and TRM acquisition for fields above $\sim 200 - 300 \mu\text{T}$ (Dunlop and Argyle, 1997; Lerner et al. 2017). TRM and ARM acquisition efficiencies are also influenced by grain size and domain state (Dunlop and Argyle, 1997); the larger grain sizes and vortex states in natural dusty olivine will result in a different calibration constant. Such variation could result in the paleointensity estimated to fall outside of the reported error range; a paleointensity estimation made with a full understanding of the magnetic domain states in the sample will be accurate.

Nevertheless, we have modified the manuscript to shift the focus from a debate over previous paleointensity estimates to furthering our understanding of magnetic vortices and achieving more accurate estimation methods that will result in the best understanding of past magnetic conditions:

Line 152-165: "Paleomagnetic data are some of the only sources of evidence of early Solar System conditions that constrain mechanisms of heating and momentum transport in the protoplanetary disk^{1,2,13-16}. Our observations and calculations show that SV or multi-vortex magnetization state Fe grains in dusty olivine will carry

magnetic remanence originating from the early Solar System. Most current paleointensity protocols implicitly assume that the magnetization carriers behave like uniform SD magnetization states, as the protocols are based on Néel's theory of SD grains⁸. Non-uniform magnetization states are the most abundant state of magnetization present in rocks and meteorites, however their thermal and temporal stabilities are poorly understood, and they have previously been considered to be poor magnetic recorders. This study presents a step change in our understanding of non-uniform magnetic states. It is now clear that a more comprehensive understanding of the thermomagnetic characteristics of magnetic vortex states will facilitate more sophisticated and sample-specific paleointensity estimates, which will further our understanding of how the protoplanetary disk evolved into our present-day planetary system."

In summary, we believe we have addressed now all the concerns of the reviewers and trust that the amended paper is appropriate for acceptance for publication in Nature Communications. We confirm that the content of this article has not been published or submitted for publication elsewhere.

Yours sincerely,

Jay Shah

REVIEWERS' COMMENTS:

Reviewer #1 (Remarks to the Author):

The major claims of this paper is the credibility and stability of vortex kamacites in dusty olivine as a paleomagnetic study. This is pretty important contributions to meteorite magnetic studies. This is novel and be interest to broader audience in meteoriticists and also paleomagnetists. Therefore, this material would possess a high-impact takeaway message about solar system formation. Although this manuscript is ready for publication. I would like to claim that authors need to describe the inapplicability of traditional Neel theory to vortex structure. Vortex kamacites should follow a non-Neel type of magnetic time-temperature relaxation, resulting in a significantly longer relaxation time at even lower temperatures during longer metamorphism in a parent body of Bishunpur.

Reviewer #2 (Remarks to the Author):

The authors have fully taken on board my comments from the previous rounds of review and have shifted the focus of the conclusions of their paper from using their results to debate previously published paleointensities to demonstrating the high paleomagnetic fidelity of magnetic vortices. I believe the authors have now demonstrated that magnetic vortices (which constitute a significant fraction of paleomagnetic carriers in natural samples) can preserve a remanence for many billions of years and they use this conclusion to strengthen the claims from previous studies that Fe particles in dusty olivines carry a reliable remanence that dates from the formation of our solar system. I now believe that the paper is ready for publication.

Reviewers' comments:

Reviewer #1 (Remarks to the Author):

The major claims of this paper is the credibility and stability of vortex kamacites in dusty olivine as a paleomagnetic study. This is pretty important contributions to meteorite magnetic studies. This is novel and be interest to broader audience in meteoriticists and also paleomagnetists. Therefore, this material would possess a high-impact takeaway message about solar system formation. Although this manuscript is ready for publication. I would like to claim that authors need to describe the inapplicability of traditional Neel theory to vortex structure. Vortex kamacites should follow a non-Neel type of magnetic time-temperature relaxation, resulting in a significantly longer relaxation time at even lower temperatures during longer metamorphism in a parent body of Bishunpur.

Response to Reviewer #1:

We have now added a note on the inapplicability of Neél theory to vortex magnetic states:

Line 71-75: "Retention of magnetic remanence over geological timescales, which is the underpinning hypothesis that enables paleomagnetism is only predicted for uniformly magnetized grains by Neel's single domain (SD) theory. Non-uniformly magnetized grains such as magnetic vortex states are not described by Neel's SD theory."

Reviewer #2 (Remarks to the Author):

The authors have fully taken on board my comments from the previous rounds of review and have shifted the focus of the conclusions of their paper from using their results to debate previously published paleointensities to demonstrating the high paleomagnetic fidelity of magnetic vortices. I believe the authors have now demonstrated that magnetic vortices (which constitute a significant fraction of paleomagnetic carriers in natural samples) can preserve a remanence for many billions of years and they use this conclusion to strengthen the claims from previous studies that Fe particles in dusty olivines carry a reliable remanence that dates from the formation of our solar system. I now believe that the paper is ready for publication.

Response to Reviewer #2:

We thank you for your comments.

In summary, we believe we have addressed now all the concerns of the reviewers and trust that the amended paper is appropriate for acceptance for publication in Nature Communications. We confirm that the content of this article has not been published or submitted for publication elsewhere.